# Analysis of Trends in Korean BIM Research and Technologies Using Text Mining

**Seungyeon Choo [1], Hyejin Park [2], Taehoon Kim [2] and Jihyo Seo [2,*]**

[1]   School of Architecture, Kyungpook National University, Daegu 41566, Korea; choo@knu.ac.kr
[2]   Department of Architecture, Kyungpook National University, Daegu 41566, Korea;
     gpwls3143@gmail.com (H.P.); thlouiskim@gmail.com (T.K.)
*   Correspondence: lelia004@naver.com; Tel.: +82-53-950-5593

**Abstract:** Building information modeling (BIM) has emerged as one of the key trends in the architecture, engineering, and construction (AEC) industry. As interest in BIM rapidly increased, the quantity of related literature also increased, thus, it has become important to analyze and identify the key topics and trends in BIM research over this period. Therefore, in order to analyze the research and technology trends related to BIM in Korea, we used the text-mining technique. In addition, by analyzing BIM-related papers using text-mining technology, we can analyze the patterns, main research trends, and trends in specific fields through data preprocessing, which formalizes the unstructured data of sentences, thus presenting us with a specific strategic plan for the future direction of research and technology related to BIM in Korea. In order to propose a strategic direction for future BIM-related research and technology in Korea, in this study, many researches related to BIM in Korea were collected and changes in the patterns and research trends of the BIM research periods were analyzed by dividing them it into the periods of introduction, development, and adaptation using text-mining methods and techniques, frequency analysis, and topic modeling methods. Therefore, in the future, BIM will be developed not only in the field of architecture but in all fields related to the construction industries, such as civil engineering, aviation, and shipping, and more active research will be conducted in various fields.

**Keywords:** BIM; trends; text mining; data analysis; topic analysis

## 1. Introduction

For the past decade, building information modeling (BIM) has emerged as one of the key trends in the architecture, engineering, and construction (AEC) industry. Despite the fact that computer-aided design (CAD) has become a major technology in the architectural field since the 1990s and 2000s, there has been a growing requirement for technological applications in the field of design in order to facilitate a more efficient construction process. The recent years have seen a fast development of the digital representation of buildings referred to as BIM [1]. BIM can first be defined as a digital representation of a building, an object-oriented three-dimensional model, or a repository of project information to facilitate interoperability and exchange of information with related software applications [2]. BIM has become an established paradigm for the development of enhanced project delivery practices [3].

The introduction of BIM in Korea raises the possibility of its usability as a new solution in the AEC industry. In reality, the construction industry required a breakthrough, and BIM presented a new direction for development. Reorganization and changes in the related standards and organizations had been made to lay the foundations for building and construction. As interest in BIM increased rapidly and as the quantity of related literature increased, it became important to analyze and identify key topics and trends in BIM research. Previous studies related to BIM trends were primarily subjective,

and the majority of the articles comprised qualitative analyses. There is insufficient extant research analyzing practical BIM trends through the analysis of the amount of BIM literature.

Therefore, in order to analyze the research and technology trends related to BIM in Korea, we use the text-mining technique. The text-mining technology system includes the fields of natural language processing, information extraction, visualization, databases, and machine learning. Therefore, by analyzing BIM-related papers using text-mining technology, we can analyze the patterns, main research trends, and trends in specific fields through data preprocessing that formalizes the unstructured data of sentences, thus presenting us with a specific strategic plan for the future direction of research and technology related to BIM in Korea.

## 2. Background

According to BIM research published in domestic and foreign journals, since approximately 2010, several tools used in existing designs and IFC(Industry Foundation Classes) standards for interoperability and the exchange of data with BIM tools have been studied, and the literature related to these tools has been published. On the basis of the theoretical research that was conducted during the early stages of the introduction of BIM, there has been an increase in the research on energy efficiency and energy performance simulations for energy saving and the cost optimization of buildings. There is also an increasing trend toward research on the role of BIM as a design and construction tool in virtual reality (VR) for improving safety and analyzing potential risks in the entire lifecycle of the building project. In addition, since 2013, research on space design through connection with the geographic information system (GIS) has been increasing, and the scope of BIM has widened. Furthermore, research linked to VR, augmented reality technology, the Internet of Things (IoT), and artificial intelligence, which are the main technologies of the fourth industrial revolution, is continuously increasing.

The speed of development of BIM-related technology and the emergence of new technologies are increasing. Research on the interoperability between these various BIM technologies is continuing, and research on the analysis of recent trends of BIM-related research in recent years is also increasing continuously. From the analysis of patterns and trends in BIM research, it is apparent that the interest in BIM has been increasing since 2004 when knowledge of BIM began to be disseminated, and the amount of literature on BIM increased accordingly. Thus, it is important to understand the key themes and trends in BIM research [4]. As the complexity of construction projects increases, research trends of implementation methods [5] and processes of digital technology related to BIM information communication technology and team collaboration [6], and research on the necessary policy trends between design and collaboration using BIM have also been increasing (particularly over the past five years (2013–2017)).

In recent years, construction projects and the related approaches to team collaboration have become increasingly complex. New technologies such as BIM tools, and novel processes and approaches have contributed to the design task, but team collaboration on existing construction projects still remains a challenge as do concerns regarding ownership and intellectual property, misunderstandings, and cultural differences. This can cause additional problems related to data generation, such as data loss, data discrepancies, errors, and liability for incorrect or incomplete data [7]. Nevertheless, the UK government recognized the importance of adopting the BIM collaboration approach and set the BIM adoption target for the construction industry in 2016. In Korea, the Public Procurement Service introduced BIM design in public buildings in 2010, and the Ministry of Land and Infrastructure intends to apply BIM to all social infrastructures by 2020. Although efforts have been made by various ministries in various countries around the world to promote it, there are few studies which use BIM throughout the project for quality control of construction and efficient use of information [8]. The following Table 1 analyzes the trends of international BIM researches by keywords.

**Table 1.** International research trends related to building information modeling (BIM) (ScienceDirect).

| Keywords | | 2013 | 2014 | 2015 | 2016 | 2017 | Total |
|---|---|---|---|---|---|---|---|
| **BIM TRENDS** | **Trend** | 152 | 197 | 227 | 282 | 351 | 1209 |
| | **Analysis** | 134 | 184 | 205 | 254 | 319 | 1096 |
| | **Policy** | 32 | 45 | 54 | 54 | 105 | 290 |
| | **Cooperation** | 18 | 19 | 26 | 22 | 51 | 136 |
| | **Status** | 71 | 70 | 76 | 94 | 122 | 433 |
| | **Problem** | 73 | 109 | 37 | 168 | 210 | 697 |
| | **Risk** | 55 | 76 | 99 | 125 | 171 | 526 |
| | **Limitation** | 57 | 85 | 101 | 125 | 169 | 537 |
| **Total** | | 592 | 785 | 925 | 1124 | 1498 | 4924 |

Furthermore, current studies on the risk of BIM and BIM-related technology development and applications have been conducted simultaneously with the progress of development research since the introduction of BIM [9]. In particular, the benefits and risks of implementing BIM in projects have been researched in part [10]. Studies are underway with the aim of identifying barriers to BIM-based risk management while focusing on a comprehensive overview of recent research on BIM-based risk management and the relationship between digital technology and traditional methods for risk management.

## 3. Research Methodology

On the basis of the current status of applications of BIM in domestic and foreign countries in the field of construction, we analyzed the current situation and problems of BIM introduction. On the basis of this global BIM status, and through an examination of the trends and direction of current BIM research in Korea, we intend to derive a direction for efficient future BIM research. First, the research papers were collected to analyze the current status of BIM research in Korea, and then the collected data were analyzed using text-mining techniques.

### 3.1. Data Collection

The collection of data for the research trend analysis was limited to the range of Korean citation index (KCI) literature published until 2017 in Korea with BIM as the search keyword. The search was conducted in three databases, DBpia(dbpia.co.kr), RISS(riss.kr), and KISS(kiss.kstudy.com), with the research results not related to BIM and duplicate results by sites excluded. Four hundred and fifty papers were considered.

In our analysis of the trends of these papers by year, following Figure 1, we found that a paper containing the keyword "BIM" was first published in 2004, with a steady increase in the number of such papers; however, there was a slight decrease in this trend since 2013. In this study, we notice three distinct periods: In Period A (introduction), the introduction of BIM and BIM-related research steadily increased from 2004 to 2010; in Period B (development), BIM-related research increased sharply from 2011 to 2013; and in period C (adaptation), it gradually declined from 2014 to 2017.

Nine institutes in total published more than 10 papers in the academic societies. The most widely published papers are those by the Architectural Society of Korea, Korea Institute of Construction Management, Korea CDE(Computational Design and Engineering) Society, Korean Academy of Industrial Science and Technology, Korea Institute of Construction Technology, the Design Convergence Association, Korea Spatial Information System Society, Computational Structural Engineering Institute of Korea, and South Korea Ecological Environment Building Society.

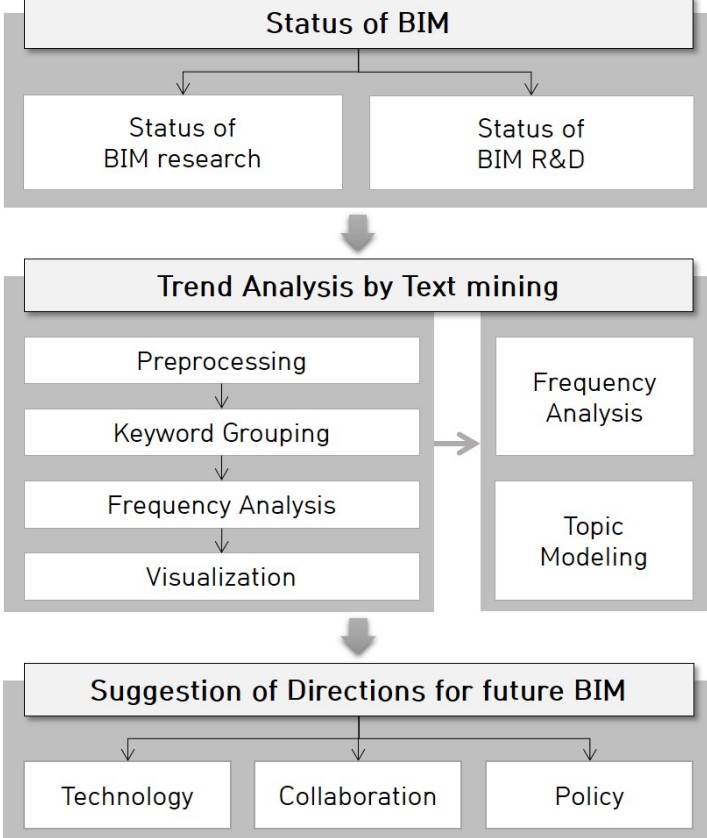

**Figure 1.** Research flow chart.

Accordign to Figure 2, Though the first paper was published in 2004, it was not until 2009 that various conferences and dissertations began to be published. In particular, following Figure 3, attempts at fusing BIM with other fields such as construction and structural development has been noticeable since 2011. Recently, studies have been undertaken in nonarchitectural fields, and not only by institutes related to information communication, such as the Korea Internet Information Society and the Korea Digital Contents Society, but also by institutes such as the Korea Railroad Society and the Korea Waterworks and Sewer Society.

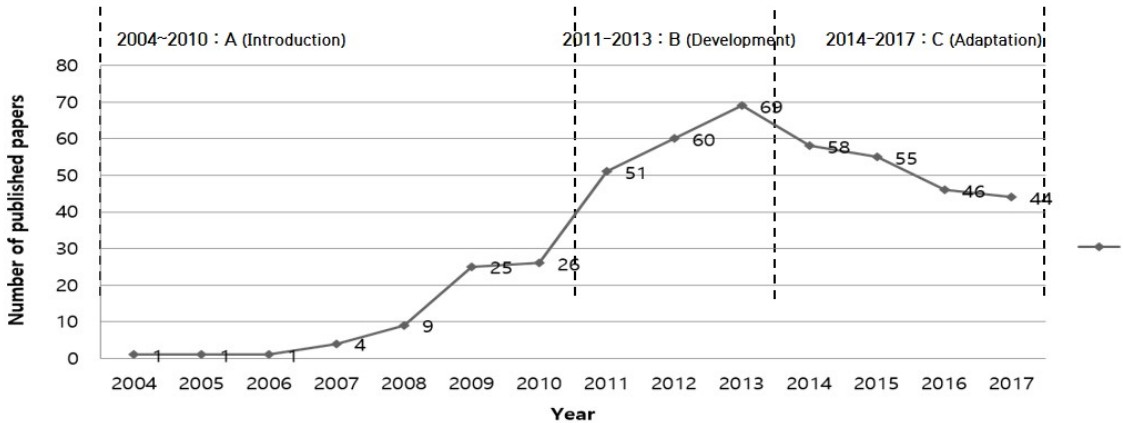

**Figure 2.** Total number of published papers per year

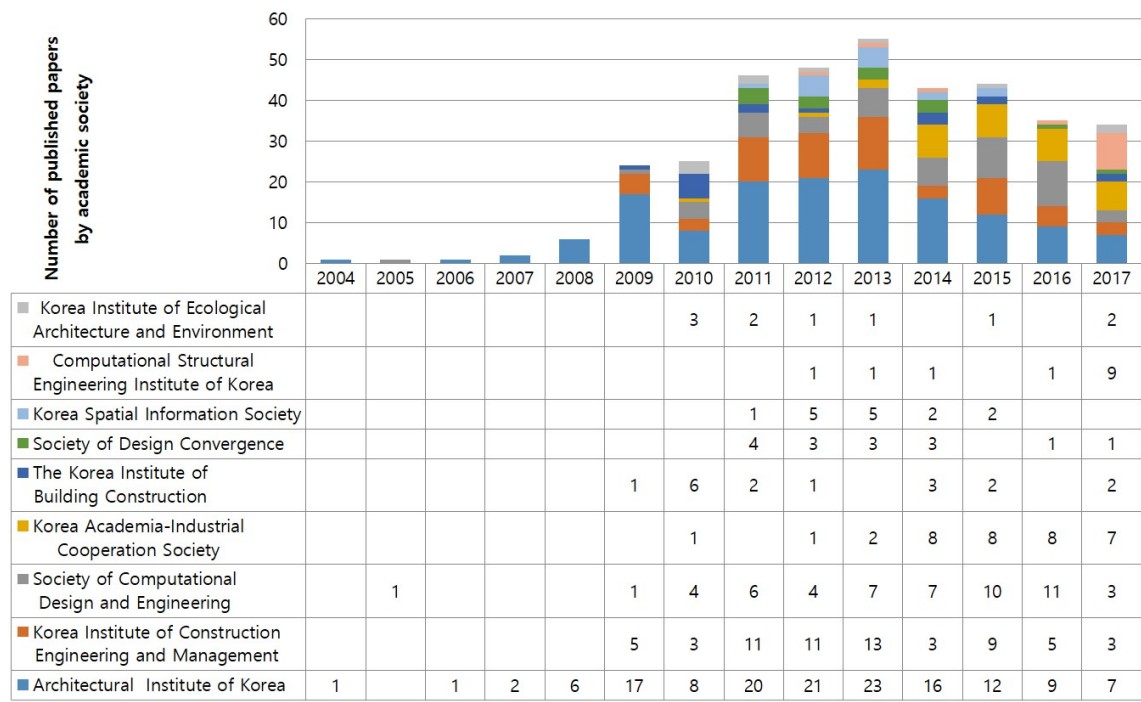

**Figure 3.** Number of papers published by academic societies

## 3.2. Data Analysis

Research and technology trends in various fields show the use of text-mining techniques. In this study, word frequency analysis and topic analysis were used. These analyses are based on the English title, keyword, and abstract; however, for more accurate results, preprocessing of the analytical data is required.

Firstly, we eliminated the numbers and special characters, then eliminated unnecessary words, e.g., articles, be verbs, conjunctions, and prepositions, to find meaningful words, and then eliminated words common to all the studies. Finally, if the word had the same meaning, it was unified with its original forms such that it was not recognized as a separate word following Table 2. Topic analysis is a probability model algorithm that extracts meaningful topics from unstructured textual materials. In this study, the latent dirichlet allocation (LDA) algorithm was used. The LDA is a generalized stochastic machine-learning model used for finding hidden topic structures in word bundles of document collections (Blei et al., 2003). It identifies potential relationships between specific words, extracts meaningful topics from the corpus, and finally displays groups with potentially similar topics.

**Table 2.** Contents of word dictionary processing.

| Step | Step1 | Step2 | Step3 | Step4 |
|---|---|---|---|---|
| Work | Delete | Delete | Delete | Replace |
| Kinds | article/ preposition/ conjunction | be Verb/Auxiliary Verb | Common Word | Original Form |
| Example | a/an/the/… in/of/for/… and/however/… | be/been/is/are/… have/must/should /… | different /analysis/study /development /aims/… | architectural architecture standards standards |

## 4. Analysis of BIM-Related Research

### *4.1. Frequency Analysis*

In the frequency analysis, excluding BIM, the keywords "building(s)", "information", and "model(ing)" all ranked high because they were searched for as BIM keywords. The other top 30 words by total counting are shown in Table 3 and by period are shown in Table 4.

**Table 3.** Frequency analysis (Total, Top 30).

| Rank | Words | Freq. | Rank | Words | Freq. | Rank | Words | Freq. |
|------|-------|-------|------|-------|-------|------|-------|-------|
| 1 | design | 953 | 11 | space | 198 | 21 | 3d | 141 |
| 2 | construction | 696 | 12 | performance | 169 | 22 | quality | 135 |
| 3 | process | 477 | 13 | standard | 167 | 23 | facility | 116 |
| 4 | system | 446 | 14 | library | 166 | 24 | evaluation | 115 |
| 5 | data | 382 | 15 | ifc | 164 | 25 | environment | 111 |
| 6 | architecture | 326 | 16 | cost | 160 | 26 | integrated | 92 |
| 7 | energy | 255 | 17 | simulation | 155 | 27 | level | 88 |
| 8 | domestic | 239 | 18 | application | 152 | 28 | software | 87 |
| 9 | structure | 238 | 19 | tool | 152 | 29 | guidelines | 84 |
| 10 | management | 234 | 20 | quantity | 144 | 30 | interoperability | 83 |

**Table 4.** Frequency analysis (Period, Top 30).

| Rank | Period A | | Period B | | Period C | |
|------|----------|-------|----------|-------|----------|-------|
| | Words | Freq. | Words | Freq. | Words | Freq. |
| 1 | design (1) | 114 | design (1) | 363 | design (1) | 476 |
| 2 | construction (1) | 125 | construction (1) | 296 | construction (1) | 275 |
| 3 | process (1) | 87 | system (1) | 191 | system (1) | 204 |
| 4 | data (1) | 53 | process (1) | 187 | process (1) | 203 |
| 5 | system (1) | 51 | data (1) | 137 | data (1) | 192 |
| 6 | architecture (1) | 48 | energy (2) | 124 | architecture (1) | 167 |
| 7 | space | 47 | architecture (1) | 111 | library (2) | 115 |
| 8 | management | 39 | management | 109 | energy (2) | 108 |
| 9 | domestic (1) | 35 | structure | 105 | domestic (1) | 105 |
| 10 | simulation (2) | 31 | domestic (1) | 99 | structure | 104 |
| 11 | structure | 29 | standard (2) | 84 | space | 87 |
| 12 | ifc (2) | 25 | quality (2) | 78 | management | 86 |
| 13 | application (2) | 24 | performance (2) | 73 | performance (2) | 85 |
| 14 | tool (2) | 24 | simulation | 66 | cost (2) | 85 |
| 15 | 3d (2) | 24 | space | 64 | ifc (2) | 80 |
| 16 | energy | 23 | application | 64 | quantity (2) | 78 |
| 17 | level (2) | 20 | quantity (3) | 63 | 3d | 76 |
| 18 | interoperability (2) | 20 | cost (3) | 60 | tool | 69 |
| 19 | standard | 19 | ifc | 59 | standard | 64 |
| 20 | traditional (2) | 19 | tool | 59 | application | 64 |
| 21 | safety | 18 | facility | 57 | evaluation | 62 |
| 22 | digital | 18 | guidelines | 55 | simulation | 58 |
| 23 | integrated | 17 | evaluation | 49 | facility | 56 |
| 24 | cost | 15 | environment | 47 | quality | 54 |
| 25 | software | 15 | software | 47 | environment | 51 |
| 26 | remodeling | 15 | library (3) | 46 | maintenance | 51 |
| 27 | exchange | 14 | 3d | 41 | engineering (3) | 48 |
| 28 | collaboration | 14 | open (3) | 41 | integrated | 40 |
| 29 | environment | 13 | green (3) | 41 | civil (3) | 40 |
| 30 | database | 13 | integrated | 35 | knowledge (3) | 40 |

(1) The top ranked words in all three periods. (2) The top ranked word for each period. (3) Newly emerging keywords for each period.

On the basis of the time frequency analysis by period, regardless of the content, the most commonly ranked words are "design", "construction", "system", "process", "data", "architecture", and "domestic." This study shows that BIM is still applied to the design phase the majority of the time, and researches related to the utilization of BIM data and studies for domestic utilization are being actively conducted. Figure 4 shows the graphs of the word frequency for each of the three periods excluding seven common words (design, construction, system, process, data, architecture, and domestic).

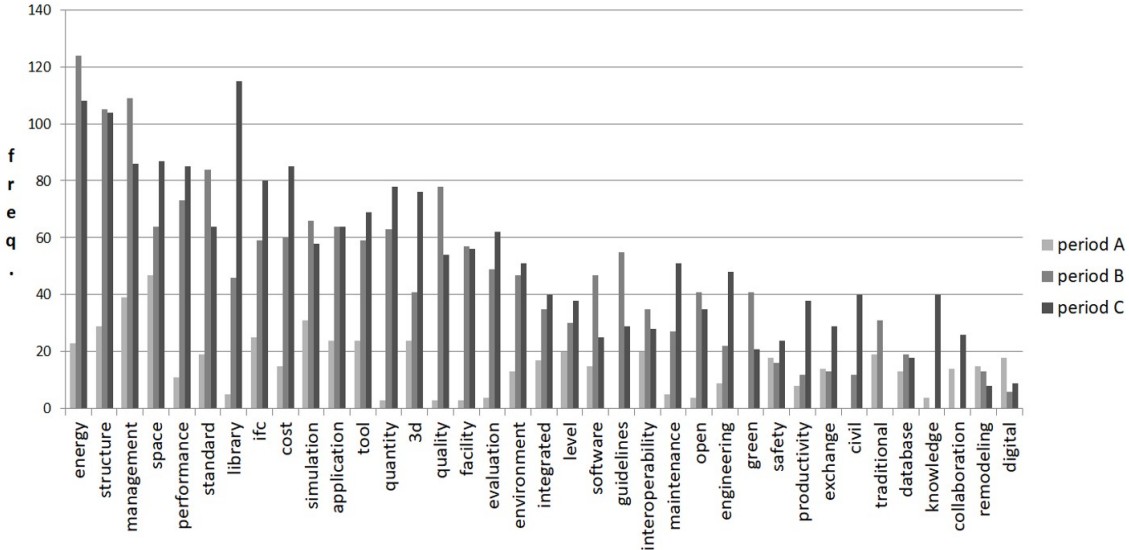

**Figure 4.** Time-frequency analysis graph

In terms of periods, the words "simulation", "ifc", "application", "tool", "3D", "level", "interoperability", and "traditional" were ranked higher in Period A than the other periods when BIM was introduced. There appears to be an interest in the problem of exploration and application of the new BIM paradigm in comparison with traditional methods and 3D authoring tools for implementing it.

In Period B (development), when BIM research was actively conducted, the use of BIM was not limited to that of just a modeling tool; as interest in the utilization of BIM increased for maintaining the quality and performance of the building, research on energy increased significantly and the word "green" emerged. In addition, words such as "quantity" and "cost" related to quantity calculation and cost analysis emerged, and research involving words such as "standard" or "guideline" was also actively conducted.

In Period C (adaptation period), following Period B, research on the performance analysis of the building in terms of energy, quantity survey, and cost was continuously performed with a noticeable interest in libraries. There appears to be a better utilization of BIM in practice. It is also interesting to note that there have been many other attempts to utilize BIM in areas other than architecture, such as railways and tunnels.

### 4.2. Topic Analysis

The topic analysis was limited to words within the top 10% of the total number of words. We extracted eight topics and 10 corresponding words for each period.

From the analysis of the introduction period of BIM from 2004 to 2010, it was found that the comprehensive contents of BIM are (Topic 1), architectural drawings and practical contents (Topic 2), safety and instruction contents (Topic 3) (Topic 4 and Topic 6), simulation and performance analysis (Topic 5), energy and remodeling construction (Topic 7), and hybrid tower and integrated forecasting (Topic 8) following Table 5.

**Table 5.** Topic analysis results of Period A (Introduction).

| | | |
|---|---|---|
| **Period A** | Topic 1 | build, model, application, projects, technology, factors, collaboration, object, factor, korea |
| | Topic 2 | bim, management, architectural, korean, work, companies, analyzed, drawing, case, progress |
| | Topic 3 | design, building, process, architectural, safety, standard, exchange, drawings, house, performed |
| | Topic 4 | interoperability, traditional, efficiency, structure, environment, tool, areas, program, cad, site |
| | Topic 5 | information, modeling, simulation, level, digital, analyze, cycle, efficient, performance, cases |
| | Topic 6 | ifc, phase, software, database, real, dynamic, essential, modernized, applications, functions |
| | Topic 7 | construction, data, project, space, energy, remodeling, results, communication, productivity, social |
| | Topic 8 | risk, buildings, integrated, form, complex, tower, visualization, aspect, estimation, introduction |

In Period B, the contents are related to performance simulations using software related to energy analysis (Topic 1 and Topic 7), designers' work or modeling quality (Topic 2 and Topic 3), work environment (Topic 4), volume, cost calculation, and standards (Topic 5) following Table 6.

**Table 6.** Topic analysis result of Period B (Development).

| | | |
|---|---|---|
| **Period B** | Topic 1 | analyzed, plans, energy, work, software, evaluation, interoperability, efficient, performed, cycle |
| | Topic 2 | estimated, designer, compared, modeled, design, modeling, developed, quality, stage, space |
| | Topic 3 | managed, application, results, projects, systems, spatial, review, public, utilization, applicability |
| | Topic 4 | data, user, class, firm, bim, management, project, technology, standard, environment |
| | Topic 5 | shapes, quantity, cost, guidelines, integrated, standards, unit, level, maintenance, framework |
| | Topic 6 | system, risk, proposed, produced, construction, process, architectural, buildings, open, facility |
| | Topic 7 | plan, draws, idm, performance, simulation, tools, library, elements, planning, technologies |
| | Topic 8 | created, enables, caused, developed, produce, named, information, building, traditional, factors |

Finally, in the period of adaptation, Period C, the contents are related to quality evaluation (Topic 1 and Topic 7), analysis of file attributes, topics related to file conversion (Topic 2), topics related to project level or work environment (Topic 3 and Topic 6), contents related to efficiency improvement such as energy and cost analysis (Topic 4), information management and utilization, and simulation related topics (Topic 5) following Table 7. Overall, during Period A, studies on the role of BIM and on the use of new software or the problems of architectural practice based on current architectural drawings were actively conducted. In Periods B and C, there was an increase in the topics of practical BIM utilization in terms of simulation, interoperability, energy performance improvement, and project management for various quantities of performance, cost calculation, and building performance analysis. This result is similar to the frequency analysis presented in Section 4.1.

**Table 7.** Topic analysis result of Period C (Adaptation).

| | | |
|---|---|---|
| **Period C** | Topic 1 | product, idm, design, architectural, quality, results, evaluation, structure, drawings, utilization |
| | Topic 2 | converter, class, measures, seed, technology, projects, phase, structural, properties, indoor |
| | Topic 3 | format, progresses, developed, quantity, project, stage, level, models, review, environmental |
| | Topic 4 | processes, introduced, energy, cost, ifc, industry, space, efficiency, architecture, civil |
| | Topic 5 | ai, ifc, information, process, modeling, model, management, application, simulation, maintenance |
| | Topic 6 | system, schema, building, performance, environment, standard, object, tools, systems, utilized |
| | Topic 7 | described, face, construction, data, library, work, buildings, assessment, open, checking |
| | Topic 8 | plan, defined, managed, engineering, spatial, knowledge, framework, facility, tool, requirements |

## 5. Conclusions

In order to propose a strategic direction for the future of BIM-related research and technology in Korea, in this study, research papers related to BIM in Korea were collected and the change patterns and research trends of the BIM research periods were analyzed by dividing them it into the periods of introduction, development, and adaptation using text-mining methods and techniques, frequency analysis, and topic modeling methods. We first conducted a keyword search on the titles of theses to know the overall trend. As a result, we realized the distribution by keyword for about 5000 papers, and it was found that the number of papers increased gradually from 2013 to 2017. In addition, we classified them by the number of articles. 'Period A' is the phase of first publication in Korea about BIM. 'Period B' is represented by a rapid increase in the number of publications. Lastly, 'Period C' is the period of a regular rate of publications. In conclusion, the introduction period of BIM includes a comparison with traditional methods in relation to the introduction of BIM, a change in the working environment, and interoperability for exploring and utilizing the new BIM paradigm. In addition, the interest in 3D authoring tools for implementing BIM implementation increased. In the development period, as a result of the increasing interest in the quality and performance of buildings, the research on energy increased significantly; BIM was not seen as being limited to functioning as a modeling tool, and research on standards and guidelines for quantity calculation, cost analysis, practical application, and project management was actively conducted. In the application of BIM in practice, various practical problems have arisen, and researches were conducted to solve these problems. As the adaptation period progressed, a steady progress in research on various technologies that could be used in BIM was observed—such as the calculation of energy, quantity surveys, and cost analysis—instead of a study of the environmental problems for initial BIM adaptation. In addition, as the necessity and importance of the literature in terms of practical applications increased, the related research increased greatly. The words 'design', 'construction', 'process', 'data', and 'system' were the most used terms in Periods A, B, and C. As time went on, we realized that words used in various fields, like civil engineering, information communication technology (ICT), among others, appeared rather than words limited to the field of architecture. The focus of attention on adaptation is a priority, and there are various efforts being made to apply BIM in areas other than construction, such as railways, tunnels, and harbors. In addition, it is being converged with advanced fields such as IoT and deep learning, which presents a potential future direction for BIM. As BIM evolves in the fourth industrial revolution, its convergence with technologies such as those associated with IoT and deep learning will result in an increase in automation, even in knowledge-based tasks. In practice, the accuracy of the simulation

of buildings will be further improved through the utilization of the accurate and diverse libraries which are available. There will be much research that calls for policy changes related to copyright problems, cost problems, and standards and guidelines, which are the greatest obstacles to the practical application of BIM.

This study analyzed the current status of Korean BIM research in the field of construction and derived trends and directions of BIM research by field from the introduction of BIM in Korea to the present. On the basis of this, we proposed a strategic direction for future BIM-related research and technology in Korea. In the future, BIM will be developed not only in the field of architecture but also in all construction industries, including civil engineering, aviation, and shipping, and more active research will be conducted in various fields.

**Author Contributions:** Conceptualization and supervision were performed by S.C., data preprocessing and Text-mining were performed by J.S., retrieve and collect data were performed by H.P., writing—original draft preperation, overall editing were performed by T.K. All the authors approved the final manuscript.

**Funding:** This research is a basic research project funded by the government (Future Creation Science Department) in 2016 with the support of the Korea Research Foundation. Assignment number: 2016R1A2B4015672.

**Conflicts of Interest:** The authors declare no conflict of interest.

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
