# Peer review of "Analysis of Trends in Korean BIM Research and Technologies Using Text Mining"

_applsci, doi:10.3390/app9204424_

Round 1

Reviewer 1 Report

The article is about a topical area in BIM research. The authors address key issues in BIM research based on a database of studies in Korea. The paper does not only propose an important brief of literature material from Korea, but also uses an interesting methodology to critically engage with BIM research. In the conclusion part, the authors try to advice qualitative findings from their analysis but it is not clear to what extent the proposed aim of research has been met. Future suggestions are where the most important contribution of the study lie in but there is no direct relation between the claims, which are discussed in Conclusion, and the research, which is clearly outlined at the end of Introduction.   The layout of the paper is pretty clear but a few suggestions can be made regarding the abstract and the introduction sections. In Abstract, the authors should give some hints about the findings of the present study and, if there may be, provide warrants about the limitations that shape their research on focus. As for Introduction, the first paragraph sits irrelevant compared to the focus of the paper. The paper talks about a current BIM research which does not need to be addressed from the 1990s' CAD research. The paper should remain focused on BIM and provide more references from this area instead of CAD research. In Background, there are strong claims that do not read academic without references. The authors should provide citations for these claims (especially in the first paragraph of Background) and reference them according to the MDPI template.   Another point to raise is the issues about the quality of English. There are repeating sentences and grammatical errors which should be corrected.   In conclusion, the paper is topical but can be more focused and detailed. It can be accepted after revision.

Author Response

Dear reviewer  

Thank you for your careful review. I corrected the points and uploaded them again.

1) This research is a study on the BIM research flow in Korea, which I belong to, and I mentioned that it was necessary to learn about CAD before BIM. In addition, before focusing on the BIM flow in Korea, we added citations on the flow of global BIM research.
2) The purpose of this study is to identify the BIM flow in Korea based on the text-mining technology that has been conducted so far, and suggest directions for future research. Therefore, the analysis obtained through the text-mining technology seems to have achieved the goal of this study, and it was possible to predict how the analysis result was fulfilled in the future.
3) Finally, we revised the abstract overall and added some hints about the findings.

After correcting the overall English grammar, the reference was corrected to the MDPI template.

Thank you for improving the quality of your research based on your advice. We will concentrate on further research in the future so that further research can be accomplished.

Thank you.

Kindly regards.

Reviewer 2 Report

This paper reviews BIM-related research in Korea, by dividing them it into the periods of introduction, development, and adaptation using text-mining methods. The topic should be of interest to the audience of the Journal of Applied Sciences. I recommend publication of the paper, but there are a few aspects that could be improved and/or explained better before the paper can be approved:

[Manuscript Abstract] Please clearly specify contributions of the research to both theory and practice.

[General] The authors need to elaborate on the generalization to the international construction sector. Currently the focus is on Korea only.

[Page 2] “…there are few studies using BIM throughout the project for quality control of construction and efficient use of information”. The issue should be explored in the broader context of literature. The citation and discussion here are insufficient; see for example https://doi.org/10.5130/AJCEB.v13i2.3120 and http://dx.doi.org/10.1016/j.autcon.2017.08.032.

[Table] Should be discussed in more details.

[Conclusion] Should highlight most significant findings.

[References] Should be amended. There are currently errors, for example: “Author2, Eissa Alreshidi, Monjur Mourshed, Yacine Rezgui, Factors for effective BIM governance, Journal of Building Engineering 2017, Vol. 10, 89–101”.      

In summary, this manuscript presents interesting results. With additional modifications, the value and quality can be further enhanced.  

Author Response

Dear reviewer

Thank you for your careful review. I corrected the points and uploaded them again.

1) This research is about the BIM research flow in Korea, which I belonged to, and I added it by quoting the flow of the world BIM research before focusing on the BIM flow in Korea.

2) The purpose of this study is to identify the BIM flow in Korea based on the text-mining technology that has been conducted so far, and suggest directions for future research. Therefore, the analysis obtained through the text-mining technology seems to have achieved the goal of this study, and it was possible to predict how the analysis result was fulfilled in the future.

After correcting the overall English grammar, the reference was corrected to the MDPI template.

Thank you for improving the quality of your research based on your advice. We will concentrate on further research in the future so that further research can be accomplished.

Thank you.

Kindly regards.

Reviewer 3 Report

Please provide the reference for “In the global architectural market, since the early 2000s, object-oriented CAD were renamed as BIM”

Many studies have identified the recent research development and trends in BIM. Why did this paper do that again for Korea only?

No eye-opening results were provided.

The references were poorly organized.

Author Response

Dear reviewer

Thank you for your careful review. I corrected the points and uploaded them again.

1) This research is a study on the BIM research flow in Korea, which I belong to, and I mentioned that it was necessary to learn about CAD before BIM. Therefore, we deleted the contents of “In the global architectural market, since the early 2000s, object-oriented CAD were renamed as BIM”. In addition, before focusing on the BIM flow in Korea, we added citations on the flow of global BIM research.

2) The purpose of this study is to grasp the BIM flow in Korea based on the text-mining technology and to suggest the direction of future research. Therefore, the analysis obtained through the text-mining technology seems to have achieved the goal of this study, and it was possible to predict how the analysis result was fulfilled in the future.

After correcting the overall English grammar, the reference was corrected to the MDPI template.

Thank you for improving the quality of your research based on your advice. We will concentrate on further research in the future so that further research can be accomplished.

Thank you.

Kindly regards.

Round 2

Reviewer 3 Report

Generally acceptable now.